# Growth Mechanism of Eutectic Si in Super-Gravity Solidified Al-Si Alloy during Annealing

**Yuehui Lu [1], Chuandong Wu [1,2,*], Hao Wu [1], Jiamin Wang [1], Yin Su [1], Zhanghua Gan [1,2,*] and Jing Liu [1,2]**

[1] State Key Laboratory of Refractories and Metallurgy, Wuhan University of Science and Technology, Wuhan 430081, China

[2] Hubei Engineering Technology Research Center of Marine Materials and Service Safety, Wuhan 430081, China

[*] Correspondence: wuchuand@wust.edu.cn (C.W.); ganzhanghua@wust.edu.cn (Z.G.)

**Abstract:** Herein, we report that the influence of annealing time on the microstructure and mechanical properties of Al-14.5Si alloys solidified under a super-gravity field. The results indicate that the coarsening of metastable eutectic Si and formation of precipitated Si could be observed at the early stage of annealing. A slight increase in yield strength and tensile strength could be observed in the sample annealed for 0.25 h, which can be ascribed to the formation of precipitated Si with limited size during the early stage of annealing. The intensified diffusion of Si atoms during annealing led to the coarsening and coalescence of the eutectic Si, as well as the coarsening of precipitated Si with further extension of the annealing time.

**Keywords:** super-gravity solidification; Al alloys; annealing





## 1. Introduction

Al-Si alloys have attracted considerable attention as conventional structural materials in the aerospace and transportation industries, mainly because of their enhanced mechanical performance, outstanding castability, and excellent corrosion resistance [1–3]. For Al-Si eutectic alloys, the mechanical behavior was significantly affected by the size, morphology, spatial distribution, and volume fraction of eutectic Si. The refining of eutectic Si was able to further enhance the strength and plasticity [4–6].

Conventional chemical modifiers and physical fields (e.g., electromagnetic fields, ultrasonic waves, and super-gravity fields) are normally considered effective approaches to refine eutectic Si [7,8]. The refining mechanisms of eutectic Si can be ascribed to the classical nucleation and growth theory, indicating that the promotion of nucleation and the inhibition of growth are conducive to refining eutectic Si. The incorporation of modifiers (Na, La and Sr et al.) results in consumption of the effective nucleation sites (AlP) of eutectic Si via the formation of $Na_3P$ and $Sr_2P$ compounds, resulting in nucleation at a higher undercooling degree [9]. Added modifier atoms may also selectively be absorbed at the twin-plane re-entrant edge (TPRE). Since the growth of eutectic Si is always detected at the TPRE, the occupation of modifier atoms at the TPRE can retard the growth of eutectic Si [10]. In addition to chemical modifiers, ultrasonic vibration is also utilized as an effective method to refine eutectic Si, mainly via the modification of Al-Si clusters by cavitation [11]. Finer Al-Si clusters, acting as nucleation sites for eutectic structure, can be achieved at higher cooling rates, inhibiting the nucleation frequency and promoting a decrease in liquid–solid transition temperature, resulting in an increase in nucleation sites [12]. Melt flow can be detected under a magnetic field, leading to the refinement of eutectic Si [13]. Meanwhile, the coarse dendrite structure is broken down into spherical structures during electromagnetic vibration.

Refinement of solidification structures can be achieved by super-gravity solidification, especially eutectic structures, such as Al-Si [14] and Cu-Sn [15]. Previously, five mechanisms were reported. (i) Crystal rain [15]: Yang et al. found that shedding of crystal nuclei was

promoted by super gravity during the early stage of solidification in Cu-Sn alloys. An increase in nucleation rate and a decrease in crystal growth improved the grain refinement effect. (ii) Dendrite fragmentation [16]: decreased grain size can be detected under a higher super-gravity field coefficient in the equiaxed crystal region, proposing a hypothesis that enhanced convection can be observed during super-gravity solidification, resulting in the breakdown of the dendrite structure, which may provide crystal nuclei and lead to the refinement of the crystal grains. (iii) Free cooling crystals and dendrite fragmentation [17]: Chang et al. concluded that an appropriate centrifugal speed, low melt superheating, and a high solute concentration are beneficial to obtain refined equiaxed grain structure based on the simulation results of Al-Si and Al-Cu alloys. (iv) Reduced critical nucleation energy [18]: a high Gibbs free energy driving force for crystallization, affected by a high ratio crystal nucleus per unit volume under super-gravity field, may result in a reduction in critical nucleation radius and an enhancement of the nucleation rate, in turn promoting grain refinement. (v) Rapid cooling rate [19]: this method can improve the content of the equiaxed crystal zone and refine the microstructure.

As evidenced by the above discussion, the refining mechanism of the solidification structure under a super-gravity field is still a debated subject. For Al-Si eutectic alloys as a modal system, it is difficult to track the microstructural evolution of the solidification structure under a super-gravity field. However, it is possible to investigate whether the eutectic structure is stable during the following heat exposure. If super-gravity solidification is able to inhibit the growth of the eutectic phase, eutectic Si may be metastable and be able to grow during the following high-temperature exposure. Thus, the purpose of the current paper is to investigate whether heat exposure has an influence on the size and distribution of eutectic Si in super-gravity solidified bulks. In order to address the above question, super-gravity solidified bulks were annealed at a temperature of $0.3T_m$ for various holding times. The microstructural evolution, mechanical properties, and growth mechanism of Al-Si alloys (solidification under super gravity of 3000 g) during annealing were investigated.

## 2. Experimental Process

Commercially pure Al and commercially pure Si were chosen as raw materials. A mixture of Al and Si (14.5%) was melted using high-frequency induction resistance at a temperature of 740 °C for 30 min. Then, ternary sodium salt was selected as a modifier (62% NaCl + 25% NaF + 13% KCl) and placed on the bottom of a graphite tube. The re-melted Al-Si ingot was heated up to 800 °C in a graphite tube with a diameter of ~20.5 mm. Subsequently, the melt was transferred from the graphite tube to a modified centrifugal apparatus (TG1850-WS). Once the graphite tube was placed in the designated position, the centrifugal rotor was accelerated to a certain speed driven by the rotating axis. After rotating for 150 s, the final sample was collected from the graphite tube. Figure 1 shows a schematic diagram of the centrifugal device. The super-gravity solidified samples were obtained with a weight of 3000 g and designated as A-0. The samples were annealed at 237 °C ($0.3T_m$) for various durations (0.25 h, 0.5 h, 1 h, 2 h, and 4 h) and designated as A-0.25, A-0.5, A-1, A-2, sand A-4, respectively.

The microstructure was analyzed by a metallographic microscope (Zeiss, Axioplan 2 imaging). A field-emission scanning electron microscope (FESEM, Nova 400) was used to evaluate the size, morphology, and spatial distribution of eutectic Si, as well as the fracture surface of the annealed samples. The average width and length of eutectic Si was measured and calculated based on FESEM images via Image J. The Vickers hardness of all the annealed samples was tested by micro-hardness indenter (HV-100B). Tensile performance tests were conducted with a universal tensile tester (Instron 5900 series). Figure 2 shows a diagram of the tensile sample. The average yield strength, tensile strength, and strain were obtained based on the testing of three nominally identical specimens for each sample.

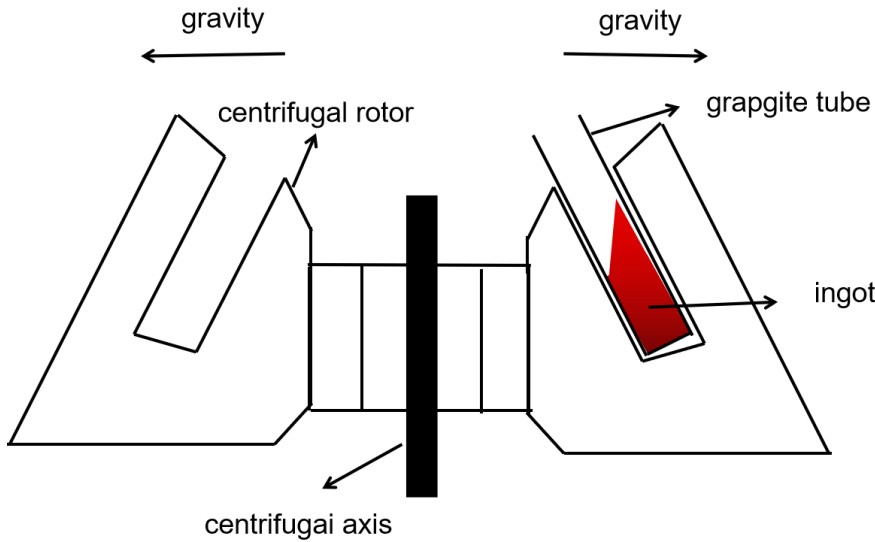

**Figure 1.** Schematic diagram of the centrifugal apparatus.

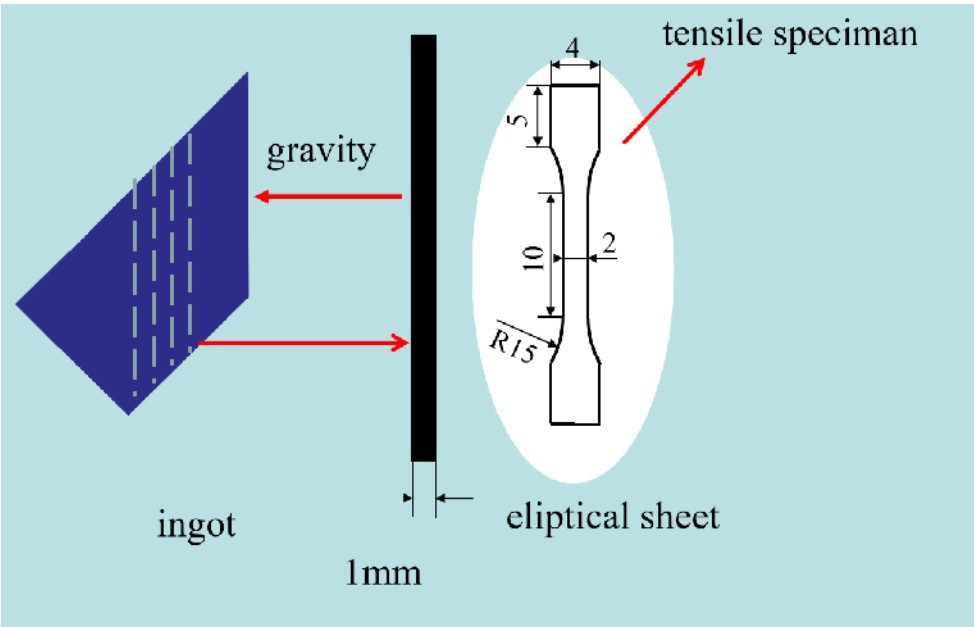

**Figure 2.** Schematic diagram of sampling of a tensile specimen.

### 3. Results

*3.1. Microstructure*

Figure 3 shows the eutectic structure of super-gravity solidified Al-Si alloys annealed for various durations. The white regions represent α-Al, while the black regions represent Al-Si eutectic. White dendrites (α-Al) grew along the axial direction in the core area. The morphology of α-Al did not change, even after annealing. Multiple dark spots could be observed in the white dendrite region after annealing, indicating that precipitation of Si (marked by yellow arrows) may have occurred during annealing. Compared with the A-0 sample, more Si could be observed in the white dendrite region with extended holding time during annealing. The eutectic structure is limited in size, making it difficult to observe with a metallographic microscope.

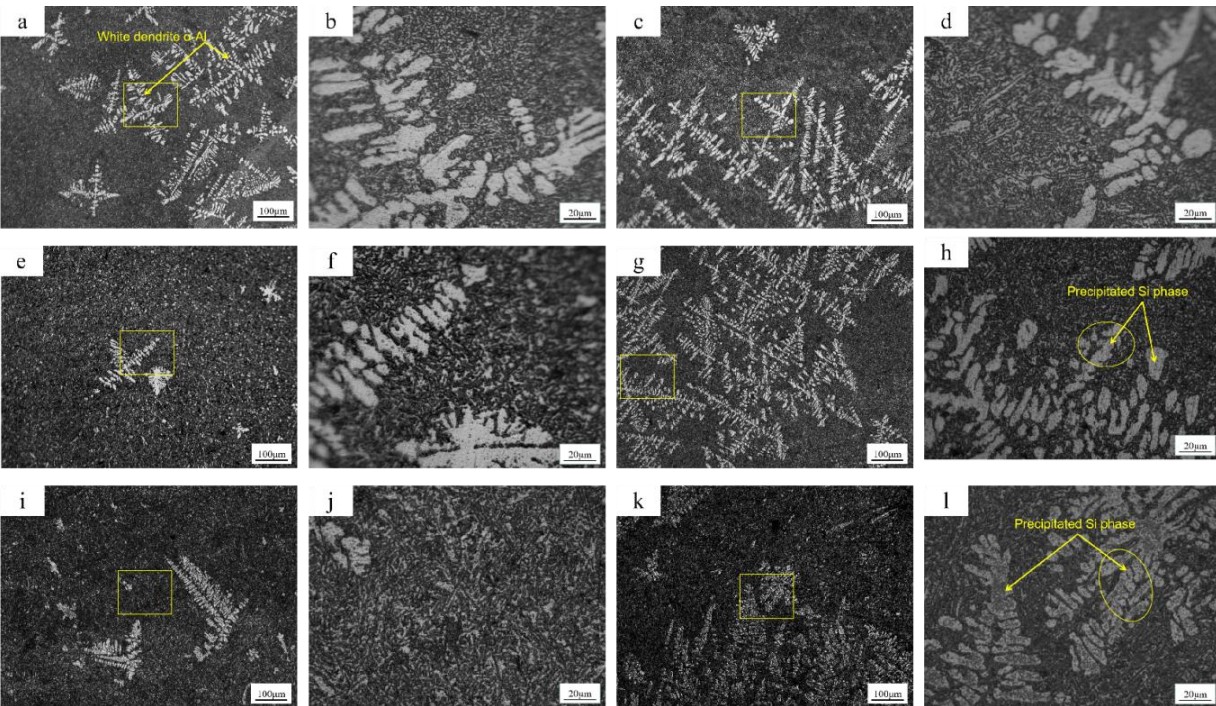

**Figure 3.** Representative OM images showing the distribution of the eutectic structure in the annealed Al-14.5Si alloys under a super-gravity field: (**a**,**b**) A-0; (**c**,**d**) A-0.25; (**e**,**f**) A-0.5; (**g**,**h**) A-1; (**i**,**j**) A-2; (**k**,**l**) A-4.

For a better understanding of the eutectic structure and precipitated Si in all the annealed Al-Si alloys, a FESEM micrograph is presented in Figure 4. A uniform distribution of short, rod-like eutectic Si (~0.47 μm in length and ~0.25 μm in width) can be observed in the A-0 sample. Coarsening of eutectic Si was detected after annealing. The average size of the A-0.25 sample is ~1.21 μm in length and ~0.34 μm in width. With further extension of the annealing time, several needle-like eutectic Si structures were observed in the A-2 and A-4 samples, which are marked by red arrows in Figure 4e,f. The average sizes of the annealed samples are provided in Table 1. Thus, the eutectic Si in super-gravity solidified A-0 sample was metastable. Coarsening and growth of this type of eutectic Si was observed after annealing. The eutectic Si particles preferred to grow along the lengthwise direction. Interestingly, the coalescence of certain rod-like Si could be observed after annealing, forming several long, needle-like Si particles with dozens of microns, as highlighted by red arrows in Figure 4e,f. Interestingly, precipitated Si was also detected after annealing.

**Table 1.** The mechanical properties of the annealed Al-14.5Si alloys solidified under a 3000 g gravity field after annealing.

| Annealing Time (h) | Yield Strength (MPa) | Tensile Strength (MPa) | Elongation (%) | Length (μM) | Width (μM) | Hardness (HV) |
|---|---|---|---|---|---|---|
| 0 h | 110.7 | 224.4 | 11.9 | 0.47 | 0.25 | 86.7 |
| 0.25 h | 121.5 | 235.9 | 12.2 | 1.21 | 0.34 | 69.3 |
| 0.5 h | 95.6 | 194.1 | 9.8 | 1.46 | 0.43 | 64.3 |
| 1 h | 105.4 | 203.5 | 9.1 | 1.82 | 0.40 | 68.7 |
| 2 h | 98.9 | 199.7 | 13.9 | 1.96 | 0.43 | 67.0 |
| 4 h | 89.9 | 179.3 | 12.8 | 2.17 | 0.56 | 64.5 |

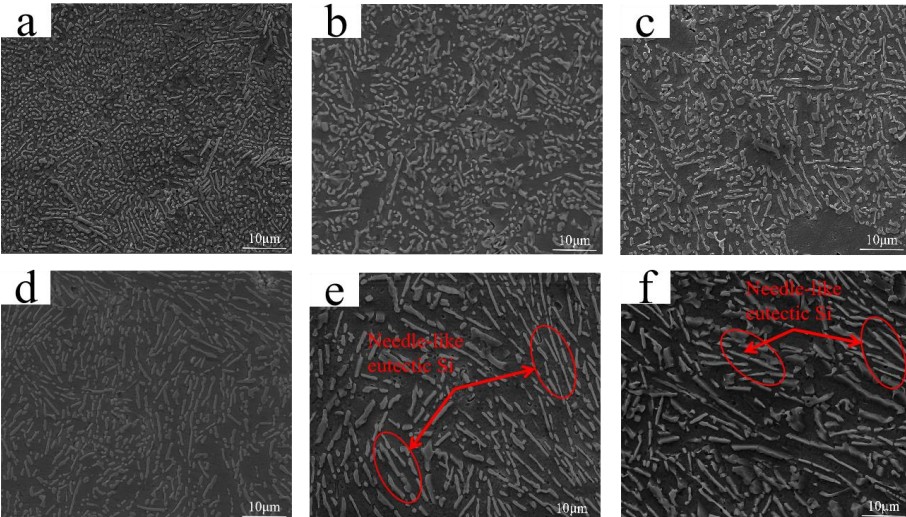

**Figure 4.** FESEM images showing the morphology and distribution of eutectic Si in the annealed Al-14.5Si alloys under a super-gravity field: (**a**) A-0; (**b**) A-0.25; (**c**) A-0.5; (**d**) A-1; (**e**) A-2; (**f**) A-4.

*3.2. Mechanical Behavior*

Figure 5 displays the Vickers hardness of the super-gravity solidified Al-14.5Si alloys annealed for various durations. The Vickers hardness of the A-0 sample (~86.7 ± 0.2 HV) is higher than that of the annealed samples. With an increasing annealing time, the Vickers hardness of the sample decreases slightly. The Vickers hardness of the A-0.5, A-1, A-2, and A-4 samples is ~64.4 ± 2.5 HV, ~68.7 ± 3.4 HV, ~67 ± 3.2 HV, and ~64.5 ± 3.3 HV, respectively (Table 1).

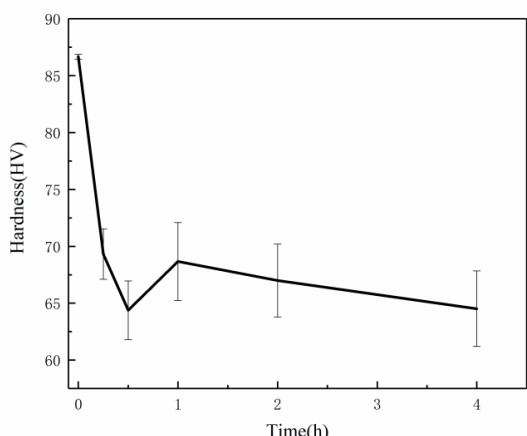

**Figure 5.** Vickers hardness of the annealed Al-14.5Si alloys solidified under a super-gravity field.

Tensile stress–strain curves of all the annealed samples are provided in Figure 6. The elongation, yield, and tensile strength of the A-0 sample are ~11.9%, ~224.4 MPa, and ~110.7 MPa, respectively. The yield (~121.5 MPa) and tensile strength (~235.9 MPa) of the A-0.25 sample are ~9.8% and ~5.1% higher than those of the A-0 sample, respectively. A slight increase in the yield and tensile strength can be observed in the A-0.25 sample, which can be ascribed to the formation of precipitated Si of limited size during the early annealing stage. For all samples except the A-0.25 sample, the yield and tensile strength of the annealed samples were lower than those of the A-0 sample. The yield (~179.3 MPa) and tensile strength (~89.9 MPa) of A-4 were ~21% and ~19% lower than those of the A-0 sample, respectively. A relatively high elongation was detected in the A-2 sample (~13.9%), which was approximately ~17% higher than that of the A-0 sample. The mechanical properties of all the samples are summarized in Table 1.

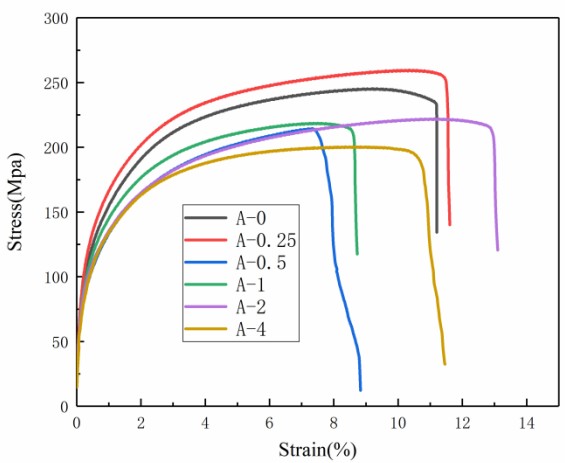

**Figure 6.** Tensile stress–strain curve of the annealed Al-14.5Si alloys solidified under a super-gravity field.

*3.3. Fractography*

Figure 7 shows the typical fracture surface of annealed Al-Si alloys (produced by a tensile test). A typical ductile transgranular fracture was found on the fracture surface, as shown in Figure 7a. However, the depth of the dimples depicted in in Figure 7b is lower than that in Figure 7a. Typically, dimples are considered to be the result of the initiation, growth, and coarsening of micropores around the second particle. Round and oval dimples of varying sizes can be observed clearly, indicating ductile fracture. The growth tendency of micropores in all directions was the same, which is typical of isometric dimples. Typically, when the dimples were deeper and larger, relatively higher elongation could be observed in the bulk. As shown in Figure 7e, the morphology and distribution of dimples in the fracture surface were homogenized, and the fracture surface was smoother after annealing for 2 h.

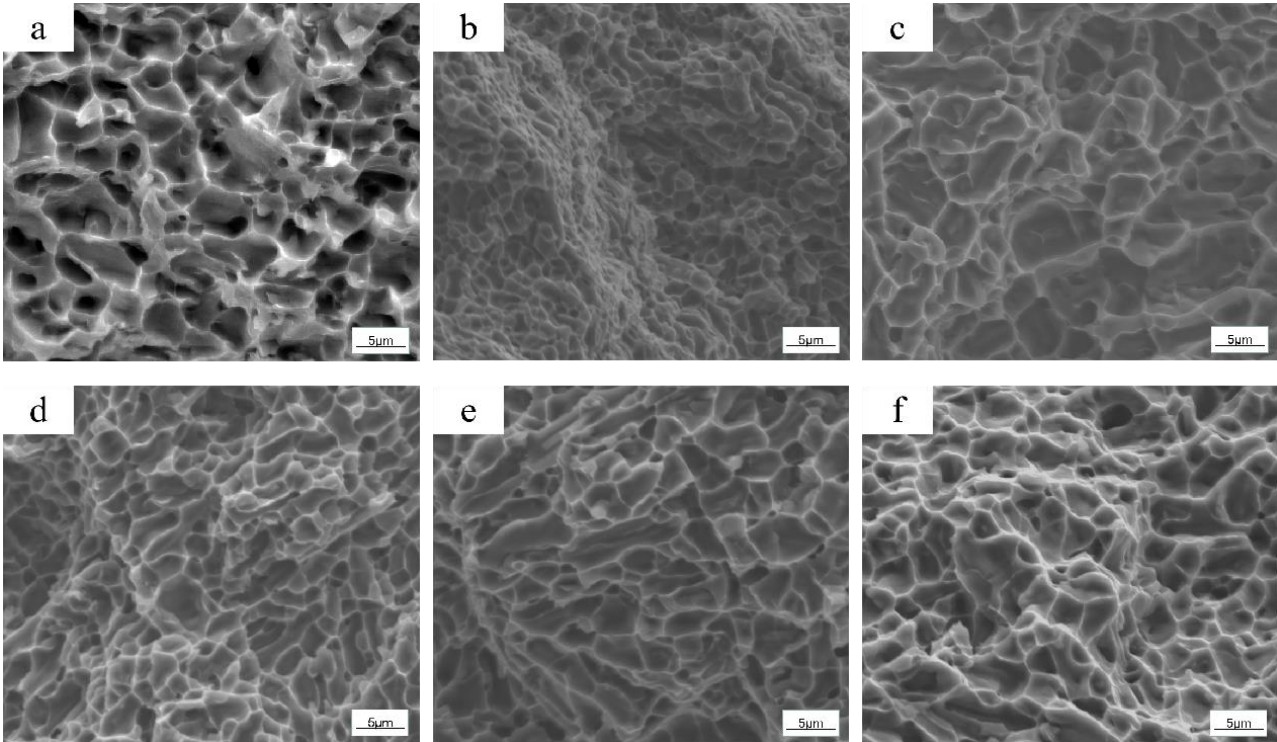

**Figure 7.** FESEM images showing the tensile fracture surface of the annealed Al-14.5Si alloys solidified under a super-gravity field: (**a**) A-0; (**b**) A-0.25; (**c**) A-0.5; (**d**) A-1; (**e**) A-2; (**f**) A-4.

## 4. Discussion

### 4.1. Refining Mechanism of Eutectic Si during Super-Gravity Solidification

It was well known that the eutectic structure can be effectively refined by super-gravity solidification [20,21]. Previous results have confirmed that the average width and length of eutectic Si decreases with an increase in the gravitational field coefficient from 1 g to 3000 g, resulting in enhancement of tensile stress and elongation of Al-Si alloys [20]. Gan et al. reported that the synergistic effect of super-gravity solidification and Co addition in refining the eutectic structure of $Al_3Ni$ [21].

In addition to this type of eutectic alloys, the refinement of pure metal or certain alloys can also be detected under the influence of super-gravity solidification. The refining of grains in super-gravity solidified pure Al was reported in [16]. Yang et al. investigated the refinement of super-gravity solidified Al-Cu and Cu-Sn alloys and clarified the refining mechanism [17,22]. In contrast to the limited solid solubility of Ni or Si in Al, the solid solubility of Cu in Al or Sn in Cu is high. The difference in composition of solids and liquids in solid solution alloys can lead to varying density during solidification. Differences in density can also be magnified during super-gravity solidification, leading to a composition gradient along the direction of the super-gravity filed. Second-phase particles can be easily isolated from the molten metal due to the enhanced difference in density [23–25]. Li et al. successfully separated iron and phosphorus phases from steel slag using a super-gravity field [23]. Lu reported that titanium albite could be effectively separated from modified Ti-bearing slag [24]. Lan reported that the separation rare earth rich phases could be achieved using a super-gravity field [25].

According to the classical nucleation and growth theory [26], the promotion of nucleation and inhibition of growth might be conductive to the refining of Si particles. A previous study reported that the gravity coefficient should be high enough (50000 g) to significantly reduce the nucleation of the solidification structure based on a calculation [27]. For the samples investigated in the present study, the density of eutectic Si was much higher than that of samples solidified under a normal gravity field. Thus, the nucleation of eutectic Si may be affected by super-gravity solidification. In addition, the refinement of eutectic Si could be observed in the super-gravity solidified sample. Previous research has confirm that $\alpha/\beta$ phases need to grow together and that the mass transfer process is an important link in the eutectic growth process for the solidification of binary eutectic alloys [28]. Super-gravity fields can significantly refine eutectic Si, which is mainly attributed to convection assisted by the super-gravity field, reducing the difference in solid–liquid composition and affecting the growth of the eutectic phase.

Figure 4 displays the coarsening of the eutectic Si phase, suggesting that the formation of eutectic Si is metastable under the influence of super-gravity solidification. Metastable eutectic Si can be also obtained by several methods, such as rapid solidification and selective laser melting. Cai et al. reported that metastable Si could be be detected in Al-27Si alloys prepared using atomization rapid solidification technology due to the extremely high cooling rate [29]. Li et al. Al-12Si alloy with a fine eutectic structure was prepared by additive manufacturing and solution treatment [30].

### 4.2. Growth Mechanism of Si

The coarsening of the eutectic Si phase in super-gravity solidified Al-Si alloys could be detected during annealing. Interestingly, certain Si particles precipitated from the $\alpha$-Al. The sources of Si atoms provided for the growth of Si can be described according to two aspects: (i) The solubility of Si in Al increased with increased temperature during the annealing process. Other Si atoms dissolved into Al during the annealing process. (ii) The limited solubility of Si (0.05 wt.%) in Al was observed in previous research [31]. In our previous work, we observed a right shift of the eutectic point in Al-14.5Si alloys solidified under a super-gravity field [20], indicating that the super-gravity field might influence the solubility of Si in the Al matrix. Supersaturated solubility of Si in Al may occur under extreme conditions, such as solidification with high-pressure treatment [32] and a rapid

cooling rate [33]. Ma et al. obtained a supersaturated solid solution of Al-20Si alloy using a high-pressure solid solution, and the amount of Si in Al reached up to 1.92% under high pressure of 1 GPa [34].

The growth and coarsening of eutectic Si and precipitated Si was mainly controlled by the diffusion of Si atoms based on the Lifshitz–Slyozov–Wagner (LSW) model [35,36], which can be expressed as follows:

$$d^n(t) - d_0^n(t) = k_d t \qquad (1)$$

where $t$ is the coarsening time, $d_0$ is the size of the eutectic Si at the onset of coarsening, and $k_d$ is the coarsening rate constant. In order to calculate the coarsening rate of eutectic Si, experimental data were input to Equation (1) ($n = 5$) for data fitting and processing to obtain Figure 8. The average coarsening rate ($k_d$, 14.7) was calculated based on the variation of eutectic Si size with annealing temperature.

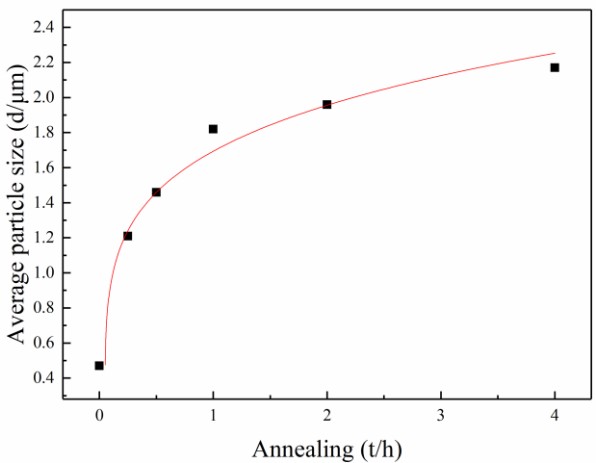

**Figure 8.** Variation of eutectic Si length with annealing time in Al-14.5Si alloy.

Due to the thermal effect of annealing and with the extension of the holding time, the diffusion rate of Si atoms further increased. The diffusion coefficient ($D$) can be expressed as:

$$D = D_0 exp^{(-Q/RT)} \qquad (2)$$

where $D_0$ is the diffusion constant, $Q$ is the activation energy per mole of atom, $R$ is the gas constant, and $T$ is the thermodynamic temperature.

The motion of Si atoms may be accelerated during annealing, resulting in coarsening and growth of eutectic Si. The thermodynamic temperature ($T$) was the main factor during diffusion process, and the annealing temperature determined atomic diffusion kinetic energy. Obvious diffusion could be observed with longer annealing time at the same temperature.

Figure 9 is a schematic diagram based on OM and SEM images, which clearly demonstrates the growth of eutectic Si and the precipitation of the Si phase on α-Al during annealing. The uniform distribution of metastable eutectic Si with a short, rod-like morphology is observed in the A-0 sample. The coarseness of the eutectic Si was observed during annealing, mainly due to mutual diffusion of Al and Si. The growth of eutectic Si occurred mainly along the lengthwise direction. Previous literature has confirmed that the preferred orientation of Si growth is along the [110] or [111] directions [37]. In addition to the coarseness of eutectic Si, precipitated Si could be observed in α-Al. Liu reported that the formation of precipitated Si was detected following high-pressure solution treatment and aging, which led to an increase in tensile stress. However, a decrease in elongation could be observed with longer aging time [32]. Thus, the formation of precipitated Si may have contributed to an increase in the tensile stress at the early stage of annealing in the A-0.25

sample. Although the most precipitated Si was observed in α-Al, it is difficult to track the formation of precipitated Si during annealing. With further extension of the annealing time, coalescence of rod-like eutectic Si could be observed after annealing, forming several long, needle-like Si particles with dozens of microns (Figure 4e,f). A slight coarsening of precipitated Si could also be observed in the annealed sample (Figure 3l). The decrease in tensile stress was mainly caused by the coalescence of eutectic Si, as well as the coarsening of the precipitated Si.

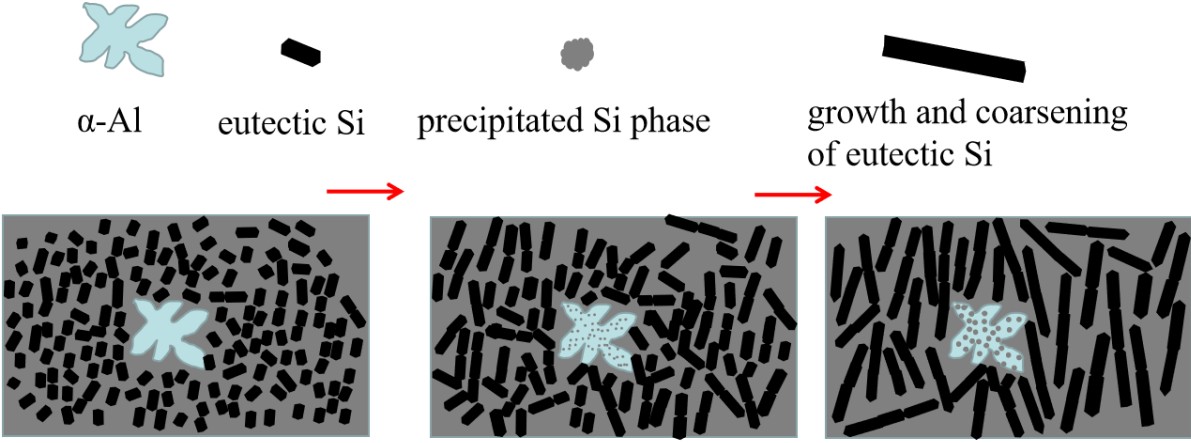

**Figure 9.** Schematic diagram of the coarsening of eutectic Si during the annealing process.

## 5. Conclusions

The influence of annealing time on the microstructure and mechanical properties of super-gravity solidified Al-Si alloys field was investigated. Our conclusions can be summarized by the following points:

(1) With an increase in annealing time, growth and coarsening of metastable eutectic Si could be observed. Interestingly, coalescence of rod-like Si could be observed after annealing, forming several long, needle-like Si particles with dozens of microns. In addition, the formation of precipitated Si was detected in α-Al, which grew during the annealing process.

(2) The yield (~121.5 MPa) and tensile strength (~235.9 MPa) of the A-0.25 sample were ~9.8% and ~5.1% higher than those of the A-0 sample, respectively. Slight increases in the yield strength and tensile strength could be observed in the A-0.25 sample, which can be ascribed to the formation of precipitated Si of limited size during the early stage of annealing. A decrease in tensile stress was observed with further extension of the annealing time.

(3) Heat exposure had an influence on the size and distribution of eutectic Si in the super-gravity solidified bulks. The mutual diffusion of Al and Si during annealing led to the coarsening and coalescence of the eutectic Si, as well as the coarsening of precipitated Si.

**Author Contributions:** Investigation, Y.L., H.W. and J.W.; Methodology, Y.S., J.L.; Supervision, C.W., Z.G.; Writing—original draft, Y.L.; Writing—review and editing, C.W., Z.G.; Funding acquistion, C.W., J.L. All authors have read and agreed to the published version of the manuscript.

**Funding:** This work was supported by the National Natural Science Foundation of China (No. 51904213) and the Natural Science Foundation of Hubei Province of China (No. 2022BEC025).

**Data Availability Statement:** Data will be made available on request.

**Acknowledgments:** This work was supported by the National Natural Science Foundation of China (No. 51904213) and the Natural Science Foundation of Hubei Province of China (No. 2022BEC025).

**Conflicts of Interest:** The authors declare no conflict of interest.

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
