# Peer review of "Growth Mechanism of Eutectic Si in Super-Gravity Solidified Al-Si Alloy during Annealing"

_crystals, doi:10.3390/cryst13040684_

Round 1

Reviewer 1 Report

The article is devoted to the study of the Si-phase growth during annealing of the Al-14.5Si alloy obtained by crystallization in a rotating mold. The experimental part represents the alloy structure at different exposure times and the effect of the formed structure on the mechanical properties of the material. The main conclusions following from the experimental part are: 1) Si particles grow during annealing; 2) an increase in the strength of the alloy is observed after aging during 0.25 hours.

Meanwhile, the title of the article contains the phrase "growth mechanism". The authors in their work only stated the fact of growth, but did not reveal the mechanism of growth. Equations 4-1 and 4-2 are given only formally, without application to the obtained experimental data.

In order for the content of the article corresponds its title, it is necessary to provide additional data on the change in the concentration of Si in α-Al and the lattice period of α-Al (an indirect indicator of the decrease in Si in α-Al) during annealing. The authors have all the experimental data in order to plot the 4-1 dependence and calculate the kd coefficient for the observed structure. It is also possible that the authors will be able to calculate the D coefficient in Equation 4-2. Having data on the change in the composition of α-Al, its period, as well as analyzing the above equations - the authors will be able to claim that they have studied the mechanism of growth of Si particles. In current version, the article is only a statement of facts.

Here are my comments on the text:

The introduction describes in detail the effect of crystallization technology under conditions of super-gravity on the structure of the alloy and the refining mechanisms. However, the main topic of the article - the growth of the phase, the stability of the microstructure after such crystallization is not given any attention.

line 103-104: "…while the black regions represent the eutectic Si". It would be more correct to say "Al-Si eutectic" here. The dark areas in the optical microscopy images in this case represent the Al-Si eutectic mixture, and not pure silicon.

Figures 3 and 4: There are a few things to note here. 1) the designations on 3h and 3l images can confuse the reader, because one might think that the arrows point to α-Al dendrites. Si precipitates inside α-Al dendrites are difficult to distinguish on such a scale. 2) The authors several times focus on Si particles inside α-Al dendrites (lines 106,214,248,253,258,267; fig. 8), but do not provide FESEM images of this Si precipitations in α-Al dendrites. They should have been added. 3) Visually, the Si particles in Figures 4a,b (A0) are larger than those in Figures 4c,d (A0.25). However, according to Table 1 and the text, the particles in the initial state are the smallest. Probably the images are mixed up. 4) Generaly, in both figures, 3 and 4, there are a lot of unnecessary images. Images with 20 µm by optical microscopy do not provide additional information; SEM images on Figure 4 with a 50 µm marker repeat the images in Figure 3 with a 100 µm marker. However, there are no important SEM images of regions inside α-Al dendrites. I dare to propose the authors to provide 3 images for each exposure time: general plan (OM 100 µm); SEM 10 µm in the region of Al-Si eutectic; SEM 10 µm inside α-Al dendrites. Such a presentation would be quite exhaustive.

line 122-123: "Thus, the eutectic Si in super-gravity solidified A-0 sample was metastable". The fact that the phase grows does not mean that it is metastable. The metastable phase is characterized by the inconstancy of the composition and crystal lattice under long-term external action, in particular, temperature exposure. It is more correct in this case to speak of a supersaturated α-Al solid solution with a nonequilibrium silicon content.

line 145-146: ... "In addition to A-0.25 sample..." The authors probably meant "with exception of A-0.25 sample", meaning "all samples except A-0.25 sample"

line 148: "A relatively high elongation could be detected in A-1 (~13.9%)…". This is inconsistent with Table 1 and the graphs in Fig. 6. Probably, the authors meant sample A-2. In general, I think that this paragraph is redundant, and only duplicates the data from Table 1 and Figure 6. It is necessary to replace this paragraph with one generalizing sentence.

Fig. 7. The images for the initial state of A-0 is not shown. In general, in this figure would be enough to show one picture with a scale of 5 µm for each exposure time. Images with lower magnifications are redundant.

line 155-156: “With the increase of annealing time, the fracture grooves tended to be homogenized…” Let the authors correct me, but I don’t observe such a trend from the given images, as well as the grooves themselves in the presented images. Only dimples are visible in the pictures.

Section "4.1 Refining mechanism..." does not refer to the experimental part of this work, but only partially repeats the text of the introduction. The authors study structural changes during annealing, but not during crystallization!

line 218: "… could be observed based on the binary phase diagram [31]". In the mentioned work [31], the Al-Si phase diagram is not presented. It is worth reformulating the sentence or providing a link to work with the phase diagram.

line 238: The term "diffusion temperature" is not clear. Previously, for T, the authors use the well-known term "thermodynamic temperature"

line 244-245: "The coarseness of the eutectic Si was able to be observed during annealing mainly due to intensified diffusion of Si atoms ". Let's not forget that in the Al-Si binary system, the displacement of silicon atoms occurs in parallel with the diffusion of Al atoms in the opposite direction. It is more correct in this case to speak of mutual diffusion of Al and Si.

Conclusion 3: The first part of the conclusion does not apply to the experimental part of the article devoted to the study of growth during annealing.

In the present form, the conclusions only descript the experimental data, but do not carry information about the mechanism of growth of the Si phase. Presentation of additional experimental data (see earlier) and particular processing of the results would allow the authors to draw a conclusion corresponding to the title of the article.

Reviewer 2 Report

The article deals with an interesting poblem of super gravity influence of the microstructural evolution, mechanical properties and growth mechanism of Al-Si alloys. The arcticle is clear written andt all comclusions supported by the experiments. But there are somne question which muct be teken into consideration the improve the article.

1. In the Introduction the authors discussed some techniques which can improve the mictrostructure and mechanical properties of Al-Si alloy at solidification. But the problem of microstructure evolution  have been fundamentally analyzed in another scientific feild, namely, for single crystal growth. An it was demonstrated that one of the main factor for design of single crystals with perfect structure was a melt state from thermodynamic point of view in the framework of solution theory. (See for instance "Axial vibration control of melt structure of sodium nitrate in crystal
growth process http://dx.doi.org/10.1016/j.jcrysgro.2014.11.022"). It would be nice to discuss this problem fo Al-Si melt.

2. The figure of the P-T-x phase diagram of Al-Si system will help the readers better understand the problem of phase transformation at cooling. The gravity force is one more intensive parameter which could change the topology of P-T-x diagram. Some drawings in this direction will be useful.

3. Is Fig.8 results from numerical simulation? Or it just a general assumption. Please, write the clear comment in the text.

4. Why Fig 4 is placed beween Fig.7 and Fig.8?

Round 2

Reviewer 1 Report

Most of the errors and inaccuracies have been fixed. The question of Si-phase metastability is debatable, but let it be left to the discretion of the authors and readers. I am grateful to the authors for taking into account many of my comments and significantly improving the perception of the article.